# Methionine Restriction Attenuates Scar Formation in Fibroblasts Derived from Patients with Post-Burn Hypertrophic Scar

**DOI:** 10.3390/ijms26125876

**Published:** 2025-06-19

**Authors:** Hui Song Cui, Ya Xin Zheng, Yoon Soo Cho, Yu Mi Ro, In Suk Kwak, So Young Joo, Cheong Hoon Seo

**Affiliations:** 1Burn Institute, Department of Rehabilitation Medicine, Hangang Sacred Heart Hospital, College of Medicine, Hallym University, 94-200 Yeong-deungpo-Dong, Yeongdeungpo-Ku, Seoul 07247, Republic of Korea; bioeast007@naver.com (H.S.C.); yxzheng2023@gmail.com (Y.X.Z.); nym8060@hanmail.net (Y.M.R.); 2Department of Rehabilitation Medicine, Hangang Sacred Heart Hospital, College of Medicine, Hallym University, 94-200 Yeong-deungpo-Dong, Yeongdeungpo-Ku, Seoul 07247, Republic of Korea; hamays@hanmail.net; 3Department of Anesthesiology and Pain Medicine, Hangang Sacred Heart Hospital, College of Medicine, Hallym University, Seoul 07247, Republic of Korea; 031132@hallym.or.kr

**Keywords:** post-burn hypertrophic scar, methionine restriction, fibroblast

## Abstract

Methionine restriction (MetR) is a common adjuvant treatment for cancer. However, studies of MetR have paid little attention to its potential implications for fibrosis. Hypertrophic scarring (HTS) is an abnormal fibrotic response after burn trauma that results from the excessive activation of fibroblasts. Because of the absence of a fully effective pharmacological treatment, HTS frequently causes great annoyance in patients as a common sequela of burns. To date, the effects of MetR on hypertrophic scar fibroblasts (HTSFs) remain unclear. This study aimed to investigate the anti-fibrotic effects of MetR and explore the associated alterations in signaling pathways in HTSFs. We isolated HTSFs from post-burn HTS tissues and cultured them in a specially prepared MetR medium. Cell and immunocytochemical staining images were captured using light and fluorescence microscopes, respectively. Cell proliferation was evaluated using a CellTiter-Glo Luminescent Cell Viability Assay Kit. mRNA and protein expression levels were determined using quantitative reverse transcription polymerase chain reaction and Western blotting, respectively. In HTSFs, MetR reduced cellular inflammation; downregulated multiple signaling pathways, including the TGF-β-SMAD, STAT, and AKT/mTOR pathways; and upregulated MAPKs. Furthermore, MetR arrested the cell cycle, promoted apoptosis, suppressed cell proliferation and migration, and reduced extracellular matrix protein secretion, thereby exerting multifaceted inhibitory effects on HTS. Our results demonstrated that MetR can inhibit scars’ formation and suggest that regulating methionine metabolism in the scar environment may help treat scars.

## 1. Introduction

Hypertrophic scarring (HTS) is a post-burn sequela that requires urgent resolution. In contrast to normal skin, HTS is characterized by prominent surface protrusions and an abnormal pigmentation; reduced elasticity, leading to tissue contracture and functional impairment; and associated local symptoms, such as itching and pain. All of these features significantly compromise the quality of life of the affected individuals [1]. HTS is more likely to develop when post-burn wounds extend to the reticular dermis of the skin; that is, second-degree or deeper [1]. The formation of post-burn HTS is primarily attributed to two key factors: prolonged inflammation and excess transforming growth factor-β (TGF-β1) [2]. Fibroblasts play a pivotal role in the pathological formation of hypertrophic scars [2]. During normal healing, at the end of the proliferative phase, fibroblasts initiate apoptosis in response to reduced TGF-β1 activity, thereby facilitating the clearance of excess cell populations as the wound progresses to maturity. However, elevated levels of TGF-β1 and decreased apoptosis disrupt this process [3].

TGF-β1 expression is slightly upregulated in normal scars; however, its most pronounced upregulation is observed in hypertrophic scar tissues, particularly in the deep dermal layer [4]. Furthermore, TGF-β1 expression is significantly upregulated in hypertrophic scar fibroblasts (HTSFs), compared to its levels in normal fibroblasts [5]. The downstream signaling of TGF-β1 is categorized into canonical and noncanonical pathways. In the suppressor of mothers against decapentaplegic (SMAD)-dependent pathway, TGF-β1 recruits and phosphorylates receptor-regulated SMADs (R-SMADs), which include SMAD2/3 and SMAD1/5/8. In the SMAD-independent pathway, TGF-β1 participates in signaling networks involving mitogen-activated protein kinases (MAPKs), such as c-Jun N-terminal kinase (JNK), p38, and extracellular signal-regulated kinase (ERK) [6]. Furthermore, prolonged inflammation activates downstream signaling pathways, such as the activation of the nuclear factor kappa-light-chain-enhancer of activated B cells (NF-κB) by TNFα and the activation of signal transducer and activator of transcription (STAT) by interleukin (IL)-6, both of which target fibroblasts to promote HTS’s formation [7].

In addition, previous studies have demonstrated that activated fibroblasts exist in a critical state primed for susceptibility to cytochrome-C-mediated mitochondrial outer membrane permeabilization (MOMP) and apoptosis. However, during fibrosis, these cells upregulate anti-apoptotic proteins to activate pro-survival mechanisms, thereby promoting cell survival [8].

Caloric restriction (CR) is a therapeutic strategy in which an organism experiences a sustained reduction in total caloric intake, while maintaining adequate nutrition to avoid malnutrition. Several studies have investigated the effects of CR on fibrosis and wound healing. In the aged obese rat model, CR results in reduced collagen deposition in the renal interstitium, specifically evidenced by the downregulation of mesenchymal markers, such as α-smooth muscle actin (α-SMA), vimentin, and N-cadherin, thereby alleviating fibrosis [9]. Subsequent studies have indicated that, compared with reducing calorie intake, restricting specific nutrients in the diet plays a more fundamental role in promoting health and extending the lifespan. The concept of “restriction” can be more precisely defined to emphasize proteins or specific amino acids [10]. Among these, methionine restriction (MetR) has garnered the most attention because of its role in extending the lifespan and inhibiting the development of cancer. The anti-inflammatory response elicited by MetR is significantly more pronounced compared to that induced by CR, thereby effectively alleviating systemic inflammation in patients [11]. Studies focusing on MetR in the context of wound healing or scars’ formation remain extremely scarce, and to date, no documented reports have established an association between MetR and HTS. One study reported that MetR did not affect wound healing in a flap model of diabetic mice [12].

Numerous studies have proposed potential approaches for the treatment of HTS. To date, the clinically implemented treatment options have been limited to surgical intervention and conservative management. Although a considerable number of drugs are currently in various stages of in vivo, in vitro, and clinical trials, no effective treatments have been identified to achieve scarless healing.

Here, we investigated the effects of MetR on hypertrophic scars to facilitate the development of new strategies for the treatment of HTS.

## 2. Results

### 2.1. Methionine Restriction Reduced HTSF Proliferation

Normally, fibroblasts exhibit an elongated, fusiform, triangular, or irregular morphology under an optical microscope [13]. As time progressed, in cell images captured on days 3 and 5, the cell size progressively increased, and the transverse length of the spindle-shaped cells significantly increased compared to that of the control cells (Figure 1A). In previous studies, immunofluorescence staining for vimentin demonstrated that the positivity rate of HTSFs was nearly 100% [13]. In the immunofluorescence staining performed in this experiment, vimentin was found to be abundantly expressed in the cytoplasm of both the experimental group and control group. In particular, the length of the cytoplasmic processes with a positive fluorescence was markedly increased (Figure 1B).

The cell density (Figure 1A) and activity of cell proliferation (Figure 1C) from day 1 to day 3 and day 3 to day 5 both exhibited significant decreases under MetR treatment compared to the proliferation of control cells. Upon treatment with MetR medium, the expression level of the cell cycle marker p21 showed a significant increase (Figure 1D), whereas proliferating cell nuclear antigen (PCNA) protein levels showed a significant decrease (Figure 1E) compared to their levels in control cells. The levels of phosphorylated AKT (Figure 1F) and mammalian target of rapamycin (mTOR) (Figure 1G) both significantly decreased after MetR. Contrary to expectations, phosphorylated p70S6K levels significantly increased under MetR treatment compared to its levels in the control group (Figure 1H).

### 2.2. Methionine Restriction Reduced Inflammation in HTSFs

Inflammation in HTSFs was significantly downregulated under MetR treatment compared to that under the control condition, as evidenced by a reduced phosphorylation of inflammatory markers, including NF-κB (Figure 2A,B), STAT1 (Figure 2A,C), and STAT3 (Figure 2A,D).

### 2.3. Methionine Restriction Reduced the Expression Levels of SMAD Proteins in HTSFs

The experimental group treated with MetR medium showed significantly downregulated levels of phosphorylated SMAD2 (Figure 2E,F) and SMAD3 (Figure 2E,G) compared to that seen in the control group treated with normal medium.

### 2.4. Methionine Restriction Reduced MAPKs in HTSFs

Contrary to the trend observed in SMAD signaling, the experimental group treated with MetR medium showed significantly upregulated levels of phosphorylated JNK1 (Figure 2H,I), p38 (Figure 2H,J), and ERK1/2 (Figure 2H,K) compared to those seen in the control group treated with normal medium.

### 2.5. Methionine Restriction Increased Apoptosis of HTSFs

The expression levels of the anti-apoptotic protein BCL2 (Figure 3A,B) showed significant decreases, and the pro-apoptotic proteins BAD (Figure 3C,D), BID (Figure 3E,F), and BAX (Figure 3G,H) showed significant increases at both the mRNA and protein levels in HTSFs treated with MetR medium compared to their levels in the control group.

We then examined the mRNA and protein expression of the apoptotic marker cytochrome c (CYCS) (Figure 3I,J), which were significantly increased in the MetR treatment group compared with thos in the control group. Subsequently, the levels of cleaved caspase 3 (Figure 3K) and its upstream regulator cleaved caspase 9 (Figure 3L) exhibited more significant increases in the experimental group than the control group. However, the protein expression levels of caspase-specific inhibitors c-IAP1 (Figure 3M) and c-IAP2 (Figure 3N) were also significantly elevated in the MetR treatment group compared to those in the control group.

Overall, the aforementioned results indicated that MetR treatment significantly induced the apoptosis of HTSFs.

### 2.6. Methionine Restriction Reduced the Expression Levels of Mesenchymal Markers in HTSFs

Previous studies have shown that fibroblasts exhibit an epithelial–mesenchymal transition (EMT)-like morphology [14]. Our experiments showed that the protein expression levels of EMT-related transcription factors (EMT-TFs), including SNAIL1 (Figure 4D), SLUG (Figure 4E), and TWIST1 (Figure 4F), showed significant decreases under MetR treatment compared to the control condition. However, unexpectedly, the expression levels of the cell surface markers E-cadherin (Figure 4A) and N-cadherin (Figure 4B) showed decreased expression levels under MetR treatment. More unexpectedly, the protein expression level of vimentin (Figure 4C) was significantly upregulated under MetR treatment compared to that under the control condition.

### 2.7. Methionine Restriction Reduced Expression of Fibrosis Markers in HTSFs

The mRNA and protein expression levels of fibrotic markers, including the differentiation marker α-SMA (*ACTA2*) (Figure 4G,H) and the ECM components type collagen (*COL1A1*) (Figure 4I,J), type III collagen (*COL3A1*) (Figure 4K,L), and fibronectin (*FN1*) (Figure 4M,N) were all significantly upregulated under MetR treatment compared to those under the control condition.

## 3. Discussion

During hypertrophic scars’ formation, a significant correlation has been demonstrated between increased fibroblast proliferation and enhanced TGF-β signaling [2]. In the present study, we showed that MetR induced a downward trend in phosphorylated SMAD2 and SMAD3 levels following decreased cell proliferation. In a mouse model of central nervous system tumors, MetR was shown to induce the upregulation of SMAD2 [15], which is a trend opposite to our findings. Dietary supplementation with Met effectively downregulates the expression level of TGF-β, which is consistent with our findings [16].

Murine L929 fibroblasts exhibit suppressed proliferation under MetR conditions through the suppression of the mTOR pathway [17]. MetR treatment has been shown to lead to the downregulation of mTOR and p70S6K expression levels in both in vivo and in vitro studies on mouse kidneys [18]. mTOR is a fundamental signaling molecule that integrates cellular metabolism, growth, and survival, and p70S6K is located downstream of mTOR. Tissues with HTS show a higher density of phosphorylated p70S6K-positive fibroblasts than tissue with normal scars [19].

AKT serves as the upstream effector of mTOR, which is also able to directly regulate cell cycle progression [20]. Inhibition of the AKT/mTOR/p70S6K signaling pathway induces cell cycle arrest and suppresses HTSF proliferation [21,22]. The signaling molecules downstream of AKT include cell cycle regulators, such as p21 and p27 [20]. p21 serves as a cell cycle checkpoint (R-point) and induces cell cycle arrest [23]. In the treatment of HTS, the downregulation of AKT’s phosphorylation promotes the expression of p21 and p27, thereby inhibiting the cell cycle and cell proliferation [24]. High p21 expression levels are typically associated with low levels of PCNA, which is a key factor in DNA repair and replication. This stems from the competitive binding of p21 to PCNA, which disrupts its activity [23]. The PCNA-positivity rate is significantly higher in tissue with HTS than in normal skin tissue [25].

In breast epithelial cells, MetR leads to a significant impairment in cell proliferation, characterized by the upregulation of p21, downregulation of cyclin D1, and subsequent cell cycle arrest [26]. Following p21’s knockdown, defects in cell proliferation induced by MetR treatment are almost entirely rescued [26]. These findings indicate that p21 plays a direct role in regulating the cell cycle during MetR.

In our results in HTSFs, the expression levels of PCNA were downregulated, while p21 levels were upregulated, and the phosphorylation levels of AKT and mTOR were reduced after MetR, which indicated suppression of the cell cycle and cell proliferation. These results are consistent with our analysis, which shows that, compared to the blank control group, both cell proliferation activity and growth density exhibit a marked reduction as the duration of MetR treatment is extended. However, the relative level of p70S6K phosphorylation was upregulated, whereas the total protein level was downregulated.

During MetR treatment in other tissues, unexpected circumstances were also observed. For instance, during a 6-h short-term MetR diet in mouse, the expression of mTOR/p70S6K signaling in liver tissues was continuously maintained at a stable level [27]. In progeria mice, a MetR diet resulted in decreased AKT phosphorylation, whereas p70S6K phosphorylation remained unchanged [28]. In a previous study on breast cancer cells, MetR increased the protein levels of p21 and p27 to induce cell cycle arrest, whereas the level of phosphorylated p70S6K remained unchanged but began to increase after 3 days of MetR treatment. The delayed activation of p70S6K has been interpreted as compensatory hyperactivation [29]. In our experiments, HTSFs were subjected to MetR for up to 5 days. Therefore, the current state may have a similar cause.

From another perspective regarding p70S6K, ERK can induce the mTOR-independent phosphorylation of p70S6K [30]. Under normal conditions, JNK1 activates p70S6K by promoting mTOR phosphorylation. When IKK2 (p50) activity is deficient, JNK1 phosphorylates p70S6K independent of mTOR, which simultaneously leads to the degradation of p70S6K [31].

NF-κB and MAPKs are two of the most representative inflammatory signaling pathways. HTS, a fibrotic disorder arising from excessive and sustained inflammatory responses, is characterized by the release of substantial amounts of reactive oxygen species (ROS) when wounds trigger proinflammatory reactions. In HTSFs, ROS activate p38, JNK [32], and NF-κB (p65) [33], thereby promoting HTSFs’ proliferation and differentiation. In chronic wound healing, inflammatory signaling pathways, such as STAT1 and STAT3, can be activated downstream of JNK, p38, and ERK [34]. The levels of phosphorylated STAT3 are significantly higher in HTSFs than in normal fibroblasts, and its inhibition may facilitate HTS treatment [35]. In HTSFs, ROS activated p38 and JNK signaling pathways, thereby promoting the proliferation and differentiation of those fibroblasts [32]. However, in studies of pathological fibrosis including HTS, MAPKs were typically investigated as components of non-SMAD signaling pathways downstream of TGF-β signaling. We have examined HTS’s formation via the SMAD and non-SMAD TGF-β signaling pathways several times. Reducing the expression of these aforementioned proteins may partially contribute to HTS treatment [36].

Under MetR treatment, we observed that the phosphorylation NF-κB, STAT1, and STAT3 were significantly downregulated in HTSFs, suggesting that MetR can effectively suppress inflammation in HTSFs. MetR exerts an inhibitory effect on the progression of chronic inflammation in a time- and dose-dependent manner [11]. MetR treatment significantly suppresses the activity, nuclear translocation, and expression of inflammatory factors downstream of NF-κB [11,15,37]. MetR treatment directly suppresses STAT3 phosphorylation [38].

However, MAPKs, including JNK, ERK, and p38, showed significantly elevated phosphorylation levels following MetR treatment. The effect of MetR on MAPK activation appears to be cell- and tissue-specific, with a general trend toward increased activation. In the mouse brain, MetR treatment significantly enhances ERK phosphorylation [39]. Phosphorylation of the p38 and JNK signaling pathways has been observed in gastric cancer cells [40]. Furthermore, the activation of MAPK signaling is often dynamically associated with the duration of treatment. In prostate cancer cells, compared to day 0, the activity of JNK1 progressively increases under the MetR condition. In contrast, the expression levels of ERK1/2 remain unchanged [41]. In liver extracts of mice, the phosphorylation of ERK1/2 rapidly increased following 5 min of MetR treatment. After 45 min of treatment, phosphorylated ERK nearly returned to control levels and remained stable thereafter. In this study, the phosphorylation of ERK was considered as an adaptive transcriptional response [27]. We propose that this is likely the reason for the MetR-induced mTOR-independent phosphorylation of p70S6K, which was compensatorily over-activated.

Further investigation of the trend of increasing MAPK levels revealed that most were associated with the promotion of apoptosis. The transient activation of JNK promotes cell survival, whereas the sustained activation of p38 and JNK induces apoptosis [42]. AKT and p38 could be activated to promote the anti-apoptotic phenotype in fibroblasts [43]. Independently, the activation of p38 and ERK is associated with resistance to apoptosis [32]. Furthermore, cell cycle arrest induced by the AKT/mTOR/p70S6K pathway can promote apoptosis upon HTS treatment [22]. Conversely, when p70S6K is studied independently, its upregulation has been shown to promote apoptosis [44]. The mechanisms by which MetR promotes apoptosis may be attributed to cell cycle arrest and inflammation driven by ROS. Under MetR conditions, the upregulation of p21 and p27 in breast cancer cells coincides with the induction of apoptosis [29].

Under MetR conditions, different types of tumor cells undergo caspase-dependent or -independent cell death [45]. MetR-induced apoptosis occurs via a mitochondrial caspase-dependent pathway. MOMP regulation leads to the release of cytochrome c from the mitochondria. Subsequently, the cascade of caspase activation, including caspase 9 and its effectors caspase 3 and 7, ultimately results in apoptosis [46]. Our analysis of signaling regulatory factors primarily focused on the Bcl-2 family and inhibitor of apoptosis (IAP) proteins. The Bcl-2 family comprises anti-apoptotic proteins, such as Bcl-2, and pro-apoptotic proteins, including Bax, BAD, and Bid, which mainly exert their effects by amplifying the MOMP signal and promoting the release of cytochrome c [47]. c-IAP1 and c-IAP2 are activated by the release of mitochondrial proteins that specifically inhibit caspase 3, 7, and 9 [48]. Upon treatment with MetR, various cancer cell lines demonstrate increased expression levels of Bcl-2 family members, including Bax and Bid, along with decreased expression level of Bcl-2 [15,45]. The majority of Bcl-2 family proteins can be regulated by JNK and/or p38 cascades [42]. For example, the activation of JNK and p38 signaling pathways led to the upregulation of apoptosis-related proteins, including Bax, Bid, and caspase 9, as well as the downregulation of Bcl-2 [49]. Both AKT and ERK contribute to the inhibition of Bad [20,50], while AKT also plays a role in suppressing caspase9 [20].

Our analysis of apoptotic markers in HTSFs revealed that MetR treatment upregulated the expression levels of the pro-apoptotic proteins Bax, Bad, and Bid and downregulated the levels of the anti-apoptotic protein Bcl-2. Subsequently, cytochrome c expression levels and cleaved caspase 3 and cleaved caspase 9 levels were markedly upregulated. In previous studies, the apoptosis of HTSFs has been harnessed for therapeutic purposes. During the inhibition of HTS’s formation, flow cytometric analysis revealed the enhanced apoptosis of HTSFs, along with markedly increased expression levels of cytochrome c and the cleavage of caspase 3 and caspase 9 [22,51]. However, the inhibitors c-IAP1 and c-IAP2 also showed increased expression levels in MetR-treated HTSFs. Given that c-IAPs can be upregulated during an increase in MOMP permeability, their elevation in our experiments is biologically plausible. Our study demonstrated that MetR acts against HTS primarily by promoting apoptosis.

In the cell images obtained in our experiments, the MetR-treated group demonstrated a significantly reduced proliferation rate and an increase in both morphology and size. However, we confirmed that MetR exerted inhibitory effects on cell proliferation, which is often accompanied by the suppression of protein synthesis and cell growth [26]. We also demonstrated p70S6K’s activation in HTSFs treated with MetR. While mTOR inhibition results in a reduction in cell size, transfection with p70S6K restores or even increases the cell size [52].

Vimentin, a key component of the cytoskeleton and an essential participant in cell adhesion, can induce the morphological transformation of cells, transforming their shape into more flattened and elongated spindle-like structures, which results in an increased cell size and cell surface area [53]. In immunofluorescence staining in the present study, fibroblasts in the experimental group retained a spindle-shaped morphology, exhibited an expanded surface area, and showed longer cytoplasmic processes with positive fluorescence. Additionally, these cells showed a significantly higher protein expression level than those in control group. Vimentin also plays a more representative role in increasing cell migration by promoting EMT [53].

EMT is primarily driven by EMT-TFs, such as SNAIL1, SLUG, and TWIST1, the expression and activation of which result from responses to various signaling pathways [54]. EMT plays a critical role in wound healing. This process is frequently observed in epithelial cells, particularly skin keratinocytes. Through the EMT process, keratinocytes contribute not only to wound repair but also to the formation of HTS [55]. Fibroblasts themselves do not typically undergo EMT, because they are already mesenchymal cells [56,57]. However, HTS has an EMT-like pathological property. Previous experimental studies have demonstrated that persistent inflammatory stimulation can induce an EMT-like morphology in fibroblasts [14]. In HTSFs, it has been shown that the expression levels of TWIST1, vimentin, and N-cadherin are significantly upregulated in comparison with HNFs, whereas the levels of the epithelial marker E-cadherin are markedly reduced, indicating an EMT-like phenotype [58].

Following MetR treatment, suppressive effects on cell migration were observed. In human gastric cancer cells, MetR treatment leads to an increase in E-cadherin expression levels, which inhibits cell invasion and metastasis [37]. Our results demonstrated that MetR significantly reduced the expression levels of TWIST1, SNAIL1, SLUG, N-cadherin, and E-cadherin. The downregulation of N-cadherin by MetR aligned with the canonical pattern of EMT inhibition, yet was accompanied by aberrant expression levels of E-cadherin. The inhibition of HTS’s formation by MetR treatment may manifest as partial antidromic EMT, with increased E-cadherin expression levels.

The primary research focus of MetR is generally related to cancer or tumors. In cancer, a dynamic form of EMT has been frequently observed, referred to as partial or hybrid EMT [59], in which cells exist in an intermediate state between epithelial and mesenchymal phenotypes. In hybrid EMT in human mammary epithelial cells, vimentin and TWIST1 levels are upregulated, whereas E-cadherin expression levels remain unchanged [60]. In fibroblasts involved in tissue repair, it was found that cells exist in a reversible activated state rather than a terminally differentiated form [61], which means a possibility of partial or hybrid EMT. Based on the aforementioned assumptions, we will further investigate the reasons for the decreased E-cadherin expression under the MetR condition. Overall, these results suggest that MetR may inhibit HTS’s formation by alleviating the EMT-like morphology of HTSFs.

The expression of vimentin and its trend under MetR treatment also did not align with the typical changes observed during EMT. However, previous studies have demonstrated that the upregulation of vimentin expression is dependent on apoptosis in palmitate- or LPS-treated human hepatocarcinoma HepG2 cells [62]. The increased expression of vimentin occurred in mice fed a methionine–choline-deficient (MCD) diet. However, non-specific caspase inhibitor treatment blocked the upregulated expression of vimentin in mice treated with LPS in combination with MCD. In our results, the upregulation of vimentin may be associated with MetR-induced apoptosis, with cleaved caspase 3 and 9 expression increased.

All of the aforementioned results collectively point to the most direct factor contributing to the suppression of HTS: fibroblast–myofibroblast trans-differentiation (FMT) and ECM overproduction. In HTSFs, fibrotic markers, such as α-SMA, type I collagen, type III collagen, and fibronectin, are expressed at significantly higher levels [58]. Our experiments demonstrated that MetR significantly reduced the expression levels of the fibroblast activation marker α-SMA and key ECM components, including typeIcollagen, type III collagen, and fibronectin. Previous studies have demonstrated that MetR downregulates the expression of genes associated with inflammation and fibrosis [63].

## 4. Materials and Methods

### 4.1. Human Specimens

HTS tissues and corresponding normal tissues were obtained from post-burn patients who provided written informed consent prior to surgery. Data on burn severity and patient characteristics are presented in Table 1.

### 4.2. Fibroblast Isolation and Culture

Fresh HTS tissue was sliced into 1 mm^3^ cubes and washed three times with ethanol and another three times with cold Dulbecco’s phosphate-buffered saline (Biowest, Riverside, MO, USA). The tissue was incubated in Dispase II solution at 4 °C overnight (1 U/mL; Gibco, Waltham, MA, USA) to separate the dermal layer, which was incubated within collagenase type IV solution (500 units/mL, Gibco) at 37 °C for 1 h. Fibroblasts were acquired supernatant after centrifugation at 1200 rpm for 5 min and cultivated in T75 flasks (Eppendorf, Hamburg, Germany) at 37 °C and 5% CO_2_, in culture medium comprising high-glucose Dulbecco’s modified Eagle’s medium (Biowest, Riverside, MO, USA) supplemented with 10% fetal bovine serum (Biowest) and 1% antibiotic–antimycotic solution containing penicillin, streptomycin, and amphotericin B (Gibco). HTSFs were cultured for 1 day until stable and were then separately cultivated in general medium (control group) and medium without L-methionine (experimental group) (Welgene, Gyeongsangbuk, Republic of Korea). The medium was replaced every 2 days. On culture day 1, 3, and 5, cell images were captured using a light microscope at ×10 magnification (IX 70; Olympus, Tokyo, Japan). HTSFs were harvested using a cell dissociation solution (Welgene) when the cell density of the control group reached 100% on day 5.

### 4.3. Immunocytochemistry (ICC) Staining

HTSFs were seeded in confocal dishes (SPL Life Sciences, Gyeonggi, Republic of Korea) and cultured in normal medium or L-methionine-free medium for 5 days. The HTSFs were fixed in acetone for 5 min and then permeabilized with PBS containing 0.25% Triton X-100 (Sigma-Aldrich, St. Louis, MO, USA) for 10 min, followed by three consecutive 5-min washes using PBS containing 0.1% Tween 20 (PBST). The cells were incubated in 10% donkey serum (Sigma) for 30 min, to block non-specific binding sites, and then with primary antibody (anti-αSMA, 1:50; Santa Cruz Biotechnology, Dallas, TX, USA) diluted in 10% donkey serum overnight at 4 °C. After three washes with PBST, the cells were incubated with the corresponding secondary antibodies (Alexa Fluor^®^ 488–conjugated; Thermo Fisher Scientific, Waltham, MA, USA) for 1 h at room temperature in the dark, followed by four additional washes with PBST. Finally, the cells were mounted using FluoroShield containing DAPI (ImmunoBioScience Corp., Davis, CA, USA) for 1 min. Cover glasses were subsequently applied and cell images were captured under a fluorescence microscope at ×10 and ×40 magnification (IX 81; Olympus).

### 4.4. Cell Proliferation Assay

HTSFs were individually seeded at densities of 2, 1, and 0.5 × 10^4^ in 96-well plates. Following 1 day of culture, the wells were divided into groups that were separately substituted with normal medium and medium without L-methionine. The time of medium replacement was considered as day 0. As a standard to determine cell viability, HSTFs with initial densities of 2, 1, and 0.5 × 10^4^ were cultivated for 1, 3, and 5 days, respectively. One hundred microliters of CellTiter-Glo reagent (CellTiter-Glo Luminescent Cell Viability Assay Kit; Promega, Madison, WI, USA) was mixed with 100 μL of medium for each well. The mixture was incubated at 37 °C for 2 h. Luminescence under excitation at 490 nm was ascertained using a DTX 880 multimode detector (Beckman Coulter, Fullerton, CA, USA), and the obtained results were computed in accordance with the following formula for the determination of cell viability: (sample luminescence − background luminescence)/(control sample luminescence − background luminescence).

### 4.5. Reverse Transcription-Quantitative Polymerase Chain Reaction (RT-qPCR)

The harvested HTSFs were supplemented with RNAzol (Cancer Rop Co., Seoul, Republic of Korea). Total RNA was extracted using the ReliaPrep RNA Miniprep System (Promega), and its concentration was measured using a NanoDrop Spectrophotometer (BioTek, Winooski, VT, USA). cDNA was synthesize using a PrimeScript RT Master Mix (Perfect Real Time; TakaRa, Shiga, Japan), and 50 ng was incorporated into each 20 µL PCR system in 96-well real-time PCR plates (Roche, Basel, Switzerland) containing 2× PCR premix (TaKaRa, Shiga, Japan), PrimeScript RT Master Mix (TaKaRa), and 0.5 µM primers (listed in Table 2). Data were obtained using a LightCycler 96 system (Roche) and processed using the 2^−ΔΔCT^ method, with GAPDH as the standard and the expression level in the control group set as 1.0.

### 4.6. Western Blotting

Harvested HTSFs were supplemented with a sufficient amount of radioimmunoprecipitation assay buffer and protease and phosphatase inhibitors (Sigma-Aldrich). After cell lysis, the proteins were extracted and quantified using a bicinchoninic acid kit (Thermo Fisher Scientific). After the addition of 5× sodium dodecyl sulfate–polyacrylamide gel electrophoresis protein-loading buffer (Cell Signaling Technology, Danvers, MA, USA), the protein samples were subjected to heating at 95 °C for 3 min for denaturation. Equal quantities of protein were added to each well, followed by electrophoresis and electroblotting onto polyvinylidene difluoride membranes. The membranes were incubated with 5% skim milk for 2 h at room temperature in a shaker. They were then incubated with primary antibody (Table 3) at 4 °C overnight. The membranes were then washed three times with 1× Tris-buffered saline with Tween-20 (TBST) and incubated for 1 h with goat anti-rabbit or goat anti-mouse horseradish–peroxidase-conjugated IgG secondary antibodies. After washing them thrice with 1× TBST, ECL luminescence solution (ATTO, Tokyo, Japan) was added to develop the protein bands, which were visualized using a chemiluminescence imaging system (WSE-6100; ATTO, Tokyo, Japan). The images were processed using ImageJ software (Version 1.53; National Institutes of Health, Bethesda, MD, USA), with the relative expression levels determined with reference to GAPDH protein expression levels, which were set at 1.0.

### 4.7. Statistical Analysis

Statistical analyses were performed using SPSS Statistics software (version 24.0; SPSS Inc., Chicago, IL, USA). The results are presented as means ± standard deviations. Further comparisons between the experimental and control groups were performed using the Mann–Whitney U test. The experimental results that demonstrated *p* < 0.05 were considered statistically significant.

## 5. Conclusions

Our results suggested that MetR is a promising strategy for HTS treatment, as it exploits the fundamental metabolic dependency of HTSFs. Its potential to inhibit HTSF growth, enhance the efficacy of conventional treatments, and impact the development of the HTS microenvironment warrants further investigation and clinical development to improve the outcomes of patients with post-burn HTS. However, further in vivo investigations will be necessary to validate these findings. Among currently established animal models, most have undergone the dietary manipulation of methionine intake, an approach that may have certain limitations. In cancer treatment, researchers have used methionine enzyme (METase) as a more convenient method to establish MetR models [37]. Furthermore, recombinant methionine enzyme (rMETase) has been developed for cancer treatment and tested in mouse models, as well as in phase I trials involving macaques and human cancer patients [64]. rMETase has demonstrated the potential to suppress cancer cell growth [65]. Moreover, Met enzyme has even been administered orally in clinical trials, where it has been shown to produce no side effects in both healthy volunteers and cancer patients [66].

## Figures and Tables

**Figure 1 ijms-26-05876-f001:**
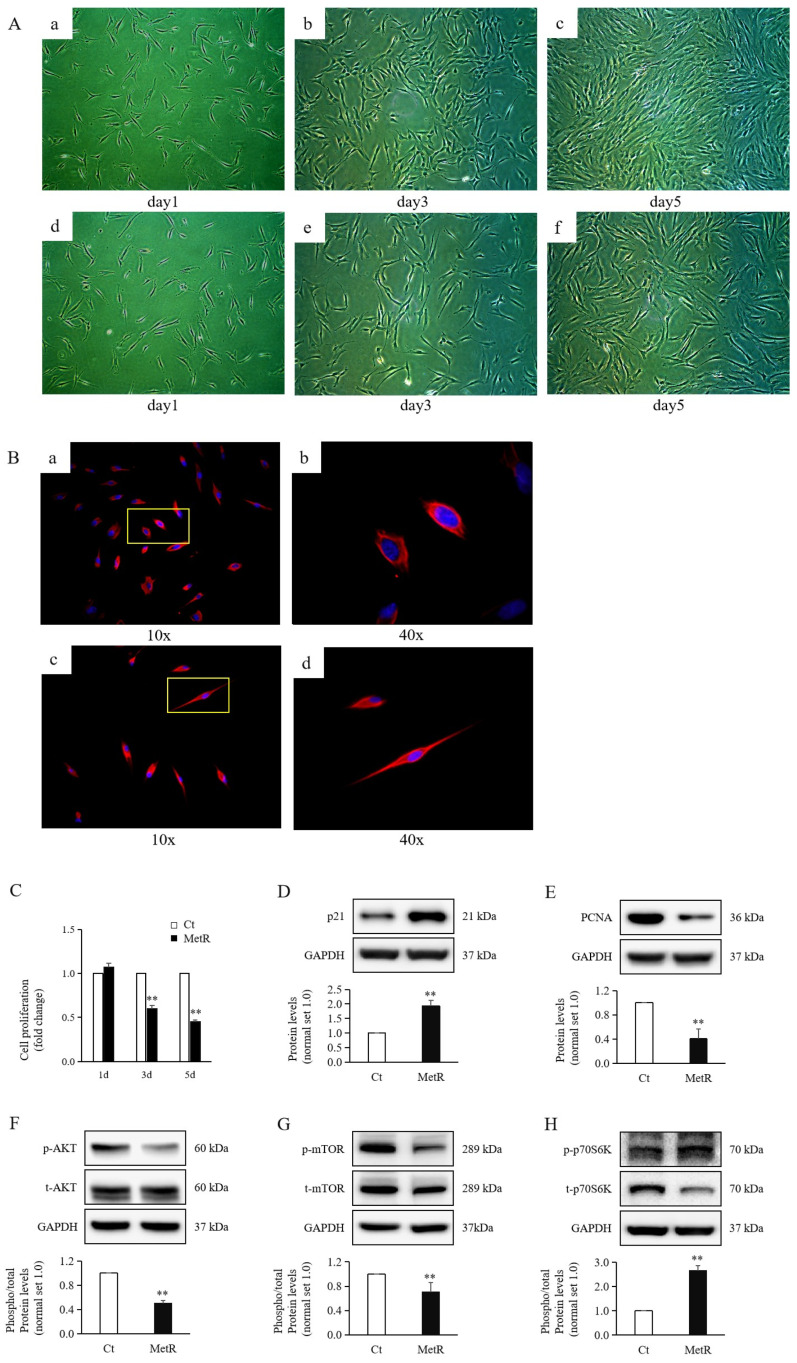
Cell morphology and cell proliferation in MetR-treated hypertrophic scar fibroblasts. (**A**) Cell images of (**a**–**c**) the control group and (**d**–**f**) the experimental group at day 1, day 3, and day 5 captured under a light microscope at ×10 magnification. (**B**) Immunofluorescence staining of vimentin in the experimental group treated with methionine restriction (MetR) medium and the control group at ((**a**): control, (**c**): MetR) ×10 and ((**b**): control, (**d**): MetR) ×40 magnification. The region delineated in the ×10 magnification image corresponds to a subset of the area captured in the ×40 magnification image. (**C**) Significantly decreased proliferation of HTSFs was observed following treatment for 3 and 5 days compared to that seen in the control groups. (**D**,**E**) Significantly increased protein levels of p21 and decreased protein levels of PCNA were observed in HTSFs treated with MetR medium compared to those in the control (Ct) cells. (**F**–**H**) Significantly decreased levels of phosphorylated AKT, mTOR, and p70S6K proteins were observed in HTSFs treated with MetR medium compared to those in the control cells. Hypertrophic scar fibroblasts (HTSFs) in each combination were extracted from the same post-burn hypertrophic scar tissues obtained from patients. HTSFs treated with normal high-glucose Dulbecco’s modified Eagle’s medium were used as the control. *n* = 3. ** *p* < 0.01 vs. control. Data represent the mean ± standard deviation (SD); *n* = 3.

**Figure 2 ijms-26-05876-f002:**
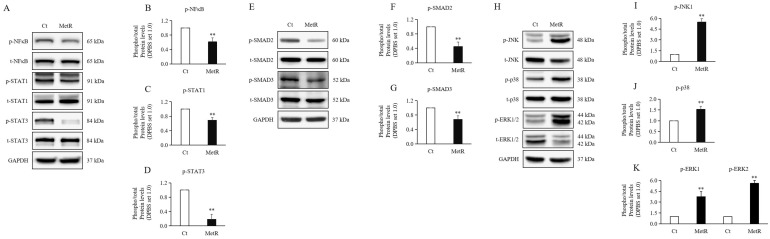
Expression of signaling pathways in MetR-treated hypertrophic scar fibroblasts. Significantly decreased levels of phosphorated (**A**,**B**) NF-κB, (**A**,**C**) STAT1, and (**A**,**D**) STAT3 proteins were observed in HTSFs treated with MetR medium compared to those seen in control cells. Significantly decreased levels of phosphorylated (**E**,**F**) SMAD2 and (**E**,**G**) SMAD3 proteins were observed in HTSFs treated with MetR medium compared to those seen in control cells. Significantly increased levels of phosphorylated (p) (**H**,**I**) JNK1, (**H**,**J**) p38, and (**H**,**K**) ERK1/2 proteins were observed in HTSFs treated with MetR medium compared to those seen in control cells. HTSFs treated with normal high-glucose Dulbecco’s modified Eagle’s medium were used as the controls. ** *p* < 0.01 vs. control. Data represent the mean ± SD; *n* = 3.

**Figure 3 ijms-26-05876-f003:**
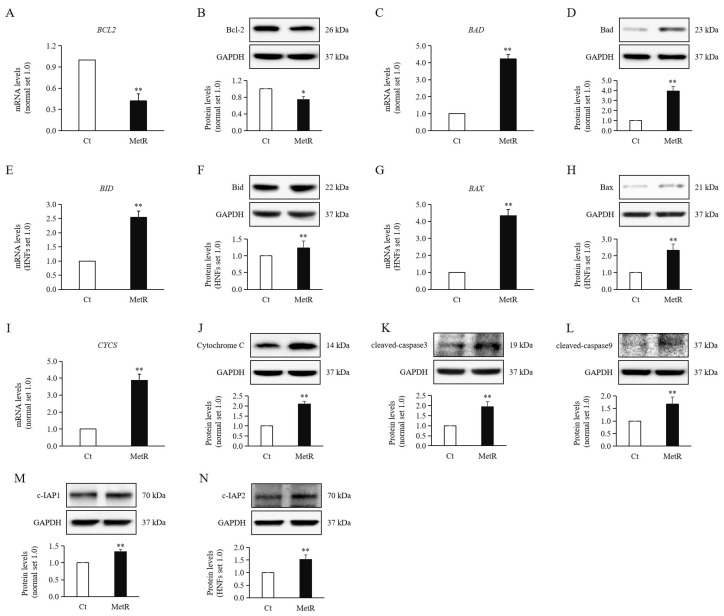
Apoptosis in MetR-treated hypertrophic scar fibroblasts. Significantly decreased mRNA and protein levels of (**A**,**B**) BCL2 and increased mRNA and protein levels of (**C**,**D**) BAD, (**E**,**F**) BID, and (**G**,**H**) BAX were observed in HTSFs treated with MetR medium compared to those seen in control cells. Significantly increased mRNA and protein levels of (**I**,**J**) cytochrome c (CYCS) were observed in HTSFs treated with MetR medium compared to those in control cells. Significantly increased protein levels of (**K**) cleaved caspase 3, (**L**) cleaved caspase 9, (**M**) c-IAP1, and (**N**) c-IAP2 were observed in HTSFs treated with MetR medium compared to those in control cells. HTSFs treated with normal high-glucose Dulbecco’s modified Eagle’s medium were used as the controls. * *p* < 0.05, ** *p* < 0.01 vs. control. Data represent the mean ± SD; *n* = 3.

**Figure 4 ijms-26-05876-f004:**
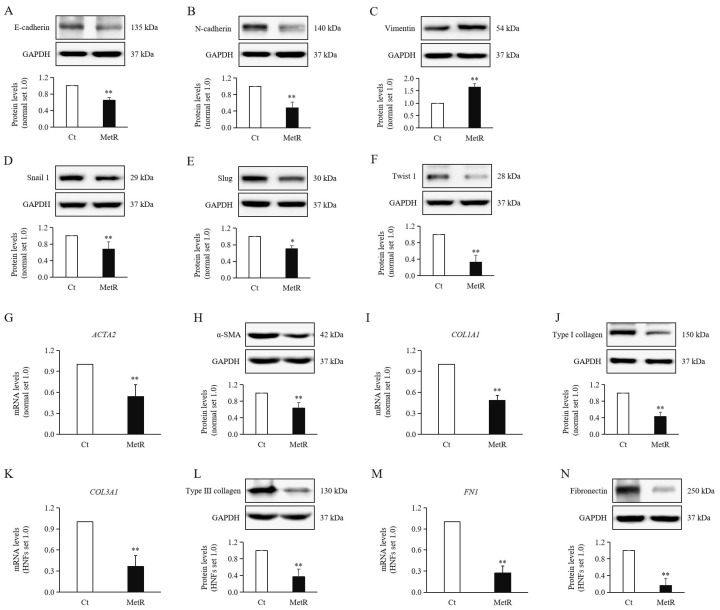
Expression of fibrosis markers in MetR-treated hypertrophic scar fibroblasts. Significantly decreased protein levels of (**A**) E-cadherin, (**B**) N-cadherin, (**D**) SNAIL1, (**E**) SLUG, and (**F**) TWIST1, and increased protein levels of (**C**) vimentin, were observed in HTSFs treated with MetR medium compared to those in control cells. Significantly decreased mRNA and protein levels of (**G**,**H**) α-SMA (ACTA2), (**I**,**J**) type I collagen (COL1A1), (**K**,**L**) type III collagen (COL3A1), and (**M**,**N**) fibronectin (FN1) were observed in HTSFs treated with MetR medium compared to those in control cells. HTSFs treated with normal high-glucose Dulbecco’s modified Eagle’s medium were used as the controls. * *p* < 0.05, ** *p* < 0.01 vs. control. Data represent the mean ± SD; *n* = 3.

**Table 1 ijms-26-05876-t001:** Demographic characteristics of patients with post-burn hypertrophic scar.

Patients	Location of Specimens	Age (Years)	Sex	Months Post-Burn
1	Foot	37	Male	9
2	Hand	30	Male	12
3	Foot	34	Male	11
4	Hand	47	Male	10

**Table 2 ijms-26-05876-t002:** Real-time polymerase chain reaction primer sequences.

Gene	Forward (5′ → 3′)	Reverse (5′ → 3′)
*CYCS*	AAGGGAGGCAAGCACAAGACTG	CTCCATCAGTGTATCCTCTCCC
*BCL2*	TGCGGCCTCTGTTTGATTT	AGGCATGTTGACTTCACTTGT
*BAD*	CCAACCTCTGGGCAGCACAGC	TTTGCCGCATCTGCGTTGCTGT
*BID*	TGGGACACTGTGAACCAGGAGT	GAGGAAGCCAAACACCAGTAGG
*BAX*	CCTTTTGCTTCAGGGTTTCA	CCATGTTACTGTCCAGTTCG
*ACTA2*	CCGACCGAATGCAGAAGGA	ACAGAGTATTTGCGCTCCGAA
*COL1A1*	ATGTTCAGCTTTGTGGACCTC	CTGTACGCAGGTGATTGGTG
*COL3A1*	CACTGGGGAATGGAGCAAAAC	ATCAGGACCACCAATGTCATAGG
*FN1*	CCAGTCCACAGCTATTCCTG	ACAACCACGGATGAGCTG
*GAPDH*	CATGAGAAGTATGACAACAGCCT	AGTCCTTCCACGATACCAAAGT

**Table 3 ijms-26-05876-t003:** Primary antibodies used in Western blotting analysis.

Target	Host	Dilution	Company (Cat. No.)
GAPDH	Rabbit	1:1000	Cell Signaling Technology (2118S), Danvers, MA, USA
GAPDH	Mouse	1:1000	Santa Cruz Technology (sc-47724), Dallas, TX, USA
α-SMA	Mouse	1:500	Abcam (ab7817), Cambridge, UK
Fibronectin	Rabbit	1:2000	Abcam (ab6328)
Collagen I	Rabbit	1:1000	Abcam (ab34710)
Collagen III	Rabbit	1:1000	Abcam (ab7778)
Vimentin	Mouse	1:3000	Abcam (ab92547)
E-Cadherin	Rabbit	1:1000	Cell Signaling Technology (3195S)
N-Cadherin	Mouse	1:1000	Thermo Fisher Scientific (333900), Waltham, MA, USA
Snail1	Rabbit	1:1000	Millipore (ABD38), Billerica, MA, USA
Slug	Rabbit	1:1000	Cell Signaling Technology (9585S)
Twist1	Mouse	1:300	Santa Cruz Biotechnology (sc-81417)
Phospho-SMAD2	Rabbit	1:1000	Cell Signaling Technology (3108S)
SMAD2	Rabbit	1:1000	Abcam (ab33875)
Phospho-SMAD3	Rabbit	1:1000	Invitrogen (MA5-14936), Carlsbad, CA, USA
SMAD3	Rabbit	1:1000	Cell Signaling Technology (9523S)
Phospho-STAT1	Rabbit	1:1000	Cusabio Technology (CSB-PA050162), Wuhan, Hubei, China
STAT1	Rabbit	1:1000	Cusabio Technology (CSB-PA825331)
Phospho-STAT3	Rabbit	1:1000	Cell Signaling Technology (9134S)
STAT3	Rabbit	1:1000	Cusabio Technology (CSB-PA004173)
Phospho-NF-κB	Rabbit	1:1000	Cell Signaling Technology (3033S)
NF-κB	Rabbit	1:1000	Abcam (ab32536)
Phospho-JNK	Rabbit	1:1000	Cell Signaling Technology (9251S)
JNK	Rabbit	1:1000	Cell Signaling Technology (9252S)
Phospho-ERK	Rabbit	1:1000	Cell Signaling Technology (4370S)
ERK	Mouse	1:1000	Cell Signaling Technology (4696S)
Phospho-p38	Mouse	1:1000	Cell Signaling Technology (9216S)
p38	Rabbit	1:1000	Cell Signaling Technology (8690S)
Phospho-AKT	Rabbit	1:1000	Cell Signaling Technology (4060S)
AKT	Rabbit	1:1000	Cell Signaling Technology (4671S)
Phospho-mTOR	Rabbit	1:1000	Cusabio Technology (CSB-PA000576)
mTOR	Rabbit	1:1000	Cusabio Technology (CSB-PA003333)
Phospho-p70S6K	Rabbit	1:1000	Cell Signaling Technology (9205S)
p70S6K	Rabbit	1:1000	Cell Signaling Technology (9202S)
p21	Rabbit	1:1000	Abcam (ab109199)
PCNA	Rabbit	1:1000	Cell Signaling Technology (2586S)
Cytochrome C	Rabbit	1:1000	Cell Signaling Technology (4272S)
c-IAP1	Mouse	1:300	Santa Cruz Biotechnology (sc-271419)
c-IAP2	Rabbit	1:1000	Cell Signaling Technology (3130S)
Cleaved caspase3	Rabbit	1:1000	Cell Signaling Technology (9661S)
Cleaved caspase9	Rabbit	1:1000	Cell Signaling Technology (9507S)
Bcl-2	Rabbit	1:1000	Abcam (ab196495)
Bad	Rabbit	1:1000	Cell Signaling Technology (9292S)
Bid	Rabbit	1:1000	Cell Signaling Technology (8762S)
Bax	Rabbit	1:1000	Abcam (ab199677)

## Data Availability

For inquiries regarding access to the datasets generated and/or analyzed in this study, please directly correspond with the corresponding author.

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
