# Peer review of "Methionine Restriction Attenuates Scar Formation in Fibroblasts Derived from Patients with Post-Burn Hypertrophic Scar"

_ijms, 2025, doi:10.3390/ijms26125876_

Round 1
Reviewer 1 Report
Comments and Suggestions for Authors
In the manuscript submitted to me for review entitled "Methionine restriction attenuates scar formation in fibroblasts derived from patients with post-burn hypertrophic scar“ the authors Hui Song Cui, Ya Xin Zheng, Yoon Soo Cho, Yu Mi Ro, In Suk Kwak, So Young Joo and Cheong Hoon Seo present a study in which they investigate the antifibrotic effects of methionine restriction (MetR) and to investigate the associated changes in signaling pathways in hypertrophic scar fibroblasts (HTSF). The presented study has a real possible future application in the treatment of scars obtained after skin burns.
The study was conducted extremely thoroughly. The purpose of the study is clearly presented. The materials and methods used are described in detail. The results are clearly presented with the help of 4 figures with multiple subfigures and are fully consistent with the conclusions made. The study used tissue samples from patients from whom written consent for inclusion in the study was obtained in advance. The conduct of the study was approved by the Ethics Committee (registration 447 number-2023-022) of Hallym University Hangang Sacred Heart Hospital, Seoul, South Korea. I cannot make any comments regarding the methods used and the results described. Everything is described in extremely detail and consistently.
My only comments are about some requirements of the journal, which are presented in the Instructions for Authors (see Instructions for Authors).
- The way of citing references in the text. They must be presented with a number, according to the order of entry, presented in brackets [ ].
- The order of arrangement of the sections - the Materials and Methods section should be after the Discussion section.
- In the References section, according to the requirements of the journal, the year, not the Volume of publication of the article, should be bolded.
- In the References section, some of the references are listed with the first 3 authors and "et al" is indicated. According to the Instructions for authors, this way of presentation is for more than 10 authors. See Instructions for authors → Reference List and Citations Guide → Download the full MDPI Reference List and Citations Style Guide:
For documents co-authored by a large number of persons (more than 10 authors), you can cite the first ten authors, then add a semicolon and add ‘et al.’ at the end:
Author 1; Author 2; Author 3; Author 4; Author 5; Author 6; Author 7; Author 8; Author 9; Author 10; et al.
Reviewer 2 Report
Comments and Suggestions for Authors
I consider this work interesting and relevant to the field of hypertrophic scar research. However, to further strengthen the manuscript and enrich the discussion of the results, I would appreciate it if the authors could answer the following specific questions:
- Regarding the increase in phosphorylated p70S6K: Although "compensatory hyperactivation" is mentioned, could the authors expand further on the potential functional implications of this increase in phosphorylated p70S6K in the context of the proliferation inhibition and other antifibrotic effects observed with MetR? Are there additional studies that support this compensatory hyperactivation, specifically in fibroblasts or under similar conditions?
- Regarding the increase in MAPKs: While a connection with p70S6K phosphorylation and a possible role in apoptosis is suggested, could the authors further discuss the potential contribution of increased JNK, ERK, and p38 to the overall antifibrotic effects observed with MetR? Could this increase have effects unrelated to apoptosis that are relevant to hypertrophic scarring?
- Regarding the EMT markers (decreased E-cadherin and N-cadherin, increased vimentin): The proposal of "partial antidromic EMT" is interesting. However, given the contradiction with the classic pattern of EMT inhibition, could the authors provide further justification or additional evidence (perhaps from other studies under similar conditions or with MetR) to support this type of partial EMT in their model? Could there be other explanations for this pattern of marker expression?
- Move to Discussion section regarding the day of decreased proliferation: Regarding Figure 1A (cell density) and Figure 1C (cell proliferation activity), once the day(s) on which significant decreases were observed under MetR treatment compared to the control group are specified in the Results section, could the authors briefly discuss in the Discussion section whether this timing coincides with or correlates with the observed changes in cell cycle markers (p21, PCNA) and signaling pathways (AKT/mTOR)?
- Limitations and Future Directions: As suggested above, it would be beneficial if the authors could include in the Conclusions section a brief mention of the most important limitations of their study (e.g., that it is an in vitro study) and propose some specific directions for future research that could validate or expand their findings (e.g., in vivo studies, exploration of more detailed molecular mechanisms, etc.).
